

# Metabolic phenotype of clinical and environmental *Mycobacterium avium* subsp. *hominissuis* isolates

Andrea Sanchini[1], Flavia Dematheis[2], Torsten Semmler[3] and Astrid Lewin[1]

[1] Division 16, Mycotic and Parasitic Agents and Mycobacteria, Robert Koch Institute, Berlin, Germany
[2] Institute of Microbiology and Epizootics, Free University Berlin, Berlin, Germany
[3] NG 1 Microbial Genomics, Robert Koch Institute, Berlin, Germany

## ABSTRACT

**Background**. *Mycobacterium avium* subsp. *hominissuis* (MAH) is an emerging opportunistic human pathogen. It can cause pulmonary infections, lymphadenitis and disseminated infections in immuno-compromised patients. In addition, MAH is widespread in the environment, since it has been isolated from water, soil or dust. In recent years, knowledge on MAH at the molecular level has increased substantially. In contrast, knowledge of the MAH metabolic phenotypes remains limited.

**Methods**. In this study, for the first time we analyzed the metabolic substrate utilization of ten MAH isolates, five from a clinical source and five from an environmental source. We used BIOLOG Phenotype Microarray[TM] technology for the analysis. This technology permits the rapid and global analysis of metabolic phenotypes.

**Results**. The ten MAH isolates tested showed different metabolic patterns pointing to high intra-species diversity. Our MAH isolates preferred to use fatty acids such as Tween, caproic, butyric and propionic acid as a carbon source, and L-cysteine as a nitrogen source. Environmental MAH isolates resulted in being more metabolically active than clinical isolates, since the former metabolized more strongly butyric acid ($p = 0.0209$) and propionic acid ($p = 0.00307$).

**Discussion**. Our study provides new insight into the metabolism of MAH. Understanding how bacteria utilize substrates during infection might help the developing of strategies to fight such infections.

Corresponding author
Astrid Lewin, LewinA@rki.de

## INTRODUCTION

*Mycobacterium avium* subsp. *hominissuis* (MAH) is clinically one of the most relevant non-tuberculous mycobacteria (*Tortoli, 2014*). MAH is an opportunistic human pathogen causing pulmonary infections, lymphadenitis in small children and disseminated infections (*Despierres et al., 2012*; *Rindi & Garzelli, 2014*). It is of increasing public health relevance, with reports of MAH infections increasing worldwide (*Hoefsloot et al., 2013*). Moreover, MAH is widespread in the environment (*Falkinham , 2013*; *Lahiri et al., 2014*). In recent years, there have been substantial advances in the analysis of bacteria at the molecular level. Indeed, several whole genome sequences are now available for many mycobacterial species, including MAH (*Bannantine et al., 2014*; *Kim et al., 2012*; *Uchiya*

*et al., 2013; Wynne et al., 2010*). In contrast, there has been little concomitant advance in knowledge at the phenotypic level. Phenotype analysis deserves greater attention, as it is the phenotype that selection pressure acts upon to confer evolutionary advantages to the bacterial species (*Plata, Henry & Vitkup, 2015*). In order to address this knowledge gap for bacterial phenotypes, BIOLOG Inc. developed the Phenotype MicroArray[TM] (PM) (BIOLOG, Hayward CA), a high throughput method for the rapid and global analysis of microbial metabolic phenotypes (*Bochner, 2003; Bochner, 2009; Bochner, Gadzinski & Panomitros, 2001; Bochner, Giovannetti & Viti, 2008*). The PM technology consists of several commercially available 96-well plates in which every well has a different substrate, allowing nearly 2,000 different microbial metabolic phenotypes to be tested (*Bochner, 2003; Bochner, 2009; Bochner, Gadzinski & Panomitros, 2001; Bochner, Giovannetti & Viti, 2008*). PM technology has been applied to several microorganisms, including mycobacteria (*Baloni et al., 2014; Bochner, Giovannetti & Viti, 2008; Borglin et al., 2012; Chen, Scaria & Chang, 2012; Gupta, Kasetty & Chatterji, 2015; Johnson et al., 2008; Khatri et al., 2013; Lofthouse et al., 2013; Mackie et al., 2014; Mishra & Daniels, 2013; Nai et al., 2013; Omsland et al., 2009; Tohsato & Mori, 2008*). One possible application of PM is the detection of phenotype changes due to gene knock-out. For example, Chen and co-authors showed that a *leuD* mutant of *M. avium* subsp *paratuberculosis* lost the ability to use several carbon, nitrogen, sulfur and phosphorous substrates (*Chen, Scaria & Chang, 2012*). Other researchers showed that the use of 12 carbon substrates differentiated *M. tuberculosis* from *M. bovis* (*Khatri et al., 2013; Lofthouse et al., 2013*).

In this study we tested clinical and environmental isolates of MAH using the PM technology. Our aim was to describe the metabolic substrates utilized by MAH isolates and to identify any metabolic differences between clinical and environmental MAH isolates.

## MATERIALS AND METHODS

### Bacterial isolates and BIOLOG phenotype microarray

We analyzed five clinical and five environmental MAH isolates (Table 1).

We performed the BIOLOG Phenotype MicroArray[TM] (BIOLOG, Hayward, CA, USA) according to the manufacturer's recommendations. The technology is based on the measurement of bacterial respiration, which produces NADH (*Bochner, Gadzinski & Panomitros, 2001*). If bacteria are able to metabolize a specific substrate, electrons from NADH reduce a tetrazolium dye in an irreversible reaction generating a purple color in the PM plate wells. This color change is measured and recorded every 15 min by the reporter instrument OmniLog[TM] (BIOLOG, Hayward, CA, USA), generating a kinetic response curve for each well (*Bochner, 2003; Bochner, 2009*).

The ten MAH isolates were tested with the 96-wells plates PM1 to PM4, containing 190 carbon (PM1 and PM2), 95 nitrogen (PM3), 59 phosphorous (PM4) and 35 sulfur (PM4) substrates. The PM plates 1, 2 and 3 include one negative control well, in which bacteria are tested without any substrate. The PM4 plate includes two negative control wells, one for the phosphorus and one for the sulfur substrates. All isolates were tested three
**Table 1  Characteristics of the ten MAH isolates analyzed in this study.**

| MAH Isolate name | Year of isolation | Source | Provider or reference | Accession of whole genome sequence |
|---|---|---|---|---|
| P-10091-06 | 2006 | Clinical—Child with lymphadenitis | NRC for Mycobacteria, Borstel, Germany | LNAV00000000 |
| 2721 | 2004 | Clinical—Child with lymphadenitis | NRC for Mycobacteria, Borstel, Germany | AWXJ00000000 |
| P-9-13 | 2013 | Clinical—Adult pulmonary infection | Charité Hospital, Berlin, Germany | LNBB00000000 |
| 104 | 1983 | Clinical—Adult pulmonary infection | Reference strain, USA | CP000479 |
| TH135 | 2013 | Clinical—Adult pulmonary infection | Reference strain, Japan | AP012555 |
| E-128 | 2010 | Environmental—Soil | Friedrich Löffler Institute, Jena, Germany | LVCS00000000 |
| E-96-2 | 2010 | Environmental—Soil | This study | LMVW00000000 |
| E- 82-7 | 2010 | Environmental—Dust | This study | LNAF00000000 |
| 27-1 | 2010 | Environmental—Dust | This study | AWXK00000000 |
| E-2514 | na | Environmental—Water | University of Düsseldorf, Germany | LNBJ00000000 |

**Notes.**

MAH, *Mycobacterium avium* subsp. *hominissuis*; NRC, National reference center; na, Not available.

times. Briefly, we cultivated each MAH isolate in 30 ml of 7H10 Middlebrook medium supplemented with 10% modified ADC-enrichment (2% of glucose, 5% of BSA, 0,85% of NaCl) until an $OD_{600\,nm}$ of 0.3–0.6 was achieved (mid-logarithmic phase of growth). The use of liquid cultures in place of agar reduces bacterial clumping. Bacterial cultures were harvested by centrifugation for 10 min at 4,000 g and pellets were re-suspended in 10 ml of distilled water. Bacterial cells were starved for one night in water at room temperature to minimize false positive reactions due to nutrient accumulation in MAH cells and to ensure the use of the substrates provided by the PM plates. The following day the cells were centrifuged and re-suspended using a sterile stick in tubes containing 10 ml of GN/GP-IF-0a (BIOLOG inoculating fluid), 120 µl of 100× BIOLOG Redox Dye Mix G and 1 ml of the appropriate additive (Table 2), until 85% transmittance was reached as measured using the turbidimeter provided by BIOLOG. In order to reduce bacterial clumping, the sterile stick used for inoculation was ground against the wall of the tube. A volume of 100 $\mu$l of this final suspension was added to each of the 96 wells of the PM plates. The PM plates were then sealed to avoid drying and incubated at 37 °C in the OmniLog® (BIOLOG, Hayward, CA, USA) incubator reader for 8 days.

As recommended by BIOLOG, we tested plates PM1 to PM4 using the same assay protocol but without the addition of bacteria in order to identify wells with abiotic dye reduction, which can generate false positive results.

## Analysis of BIOLOG phenotype microarray data

The raw kinetic data were exported as CSV files using OmniLog PM file Management/kinetic Analysis module (*Bochner, 2003*; *Khatri et al., 2013*). Differences in the metabolization of the different substrates by the ten MAH isolates were investigated by analyzing the maximum height of the bacterial respiration curves (parameter A) using the R-package opm (*Vaas et al., 2013*). To allow comparisons across plates processed in different experimental runs, the A parameters were normalized by subtracting the well mean of the negative control (*Vaas et al., 2013*). Furthermore, the A parameters of the triplicates

**Table 2 Additives used for each PM plates.** As additive are usually provided nutrient that are absent to the PM minimal media, but present in a standard MAH growth conditions. We used additives to make a complete minimal medium but omitted anything that could act as a source of the substrates of interest (for example, we did not include nitrate additives in the nitrogen source plates).

|  | Additive a | Additive b |
|---|---|---|
| PM plate usage | PM1, PM2, PM4 | PM3 |
| Ingredients | 24 mM MgCl2 | 24 mM MgCl2 |
|  | 12 mM CaCl2 | 12 mM CaCl2 |
|  | 0,0012% ZnSO4 | 0,0012% ZnSO4 |
|  | 0,06% ferric ammonium citrate | 0,01% tween 80 |
|  | 1,2% NH4Cl |  |
|  | 0,01% tween 80 |  |

were combined by calculating the mean and discretized into "positive," "moderate" and "negative" metabolization using the method "discrete" within the R-opm package. Substrates differentiating the isolates from each other were visualized as a heatmap generated using the R-packages heatmap.plus with the Euclidean algorithm. The heatmap displays the utilization of each substrate with a color key: yellow for strong positive metabolization, green for moderate metabolization and blue for no metabolization.

## Analysis of metabolic pathways

The metabolic pathways of the two substrates of interest butyric and propionic acid have been further analyzed. Specifically, we extracted all sequences of the genes known to be associated with the pathways related to butyric and propionic acid from the KEGG pathway database (*Kanehisa et al., 2016*). We extracted the genes from all the *M. avium* subspecies ($n = 8$) present in the KEGG pathway database, namely: *M. avium* subsp. *paratuberculosis* K-10, *M. avium* subsp. *paratuberculosis* MAP4, *M. avium* subsp. *paratuberculosis* E1, *M. avium* subsp. *paratuberculosis* E93, *M. avium* subsp. *avium* DJO-44271, *M. avium* subsp. *avium* 2285 (R), *M. avium* subsp. *avium* 2285 (S) and the *M. avium* 104. The redundant genes have been excluded. Then we screened all such genes in genomes of our ten MAH isolates by performing a Custom BLAST analysis using Geneious version 9 (*Kearse et al., 2012*). The parameters for the screening that we used to determine if a gene was present or not were: sequence identity $\geq 90\%$, sequence coverage $\geq 90\%$, *e* value $\leq 0.01$.

In addition, we analyzed the number of Single Nucleotide polymorphisms (SNP)s (both synonymous and nonsynonymous) in the sequence of the genes detected in our MAH isolates. For each gene, we also constructed a phylogenetic tree using the nucleotide sequences to determine whether any SNP was associated with clinical or environmental source of the isolates based on the Tamura–Nei model using Geneious version 9.

## Statistical analyses

We generated two groups, one with data from all clinical isolates and the other with data from all environmental isolates. Statistical differences between clinical and environmental isolates in the metabolization of butyric acid and propionic acid were evaluated by means of 95% family-wise comparison of group means (Tukey contrast test) of the parameter A

on specific wells using the function "opm_mcp" within the opm R-package. A *p* value less than 0.05 was considered to be statistically significant.

### Whole genome sequencing of MAH isolates

Genomic DNAs were extracted from the MAH isolates as described previously (*Lewin et al., 2003*). Whole genome sequencing (WGS) was performed using Illumina MiSeq 300 bp paired-end sequencing, yielding a coverage that exceeded $100\times$. The NGS QC tool kit was used to assess the quality of the data reads, which was set as reads with a minimum of 70% of bases having a phred score greater than 20 (*Patel & Jain, 2012*). De novo assembly of the resulting reads into multiple contigs was performed using CLC Genomics Workbench 8.0 (CLC bio, Aarhus, Denmark) and contigs annotation was done using RAST (*Aziz et al., 2008*).

### Determination of the maximum common genome and of the accessory genome

We determined the maximum common genome (MCG), comprising those genes present in all of the ten MAH genomes, as reported previously (*Von Mentzer et al., 2014*). All these genes were then extracted from all genomes, concatenated and aligned. The resulting alignment was used to generate a clustering tree using RAxML 8.1 (*Stamatakis, 2014*).

For determination of the accessory genome we applied the PanGenome Pipeline – Roary. After determination of the accessory genome of the ten MAH genomes and its distribution within them, we separated those genes that are exclusively present only in either the environmental strains or the clinical strains (*Page et al., 2015*).

## RESULTS

### Substrate utilization of the ten MAH isolates

We tested the capability of our ten MAH isolates to metabolize 379 different substrates. In total, 334/379 (88.1%) substrates were negative for all of the isolates (see Table S1). A total of 23/379 (6.1%) substrates caused abiotic reactions and were excluded from further analysis. A list of false-positive substrates is shown in the Table S2. The kinetic curves corresponding to the control plates PM1 to PM4 tested without bacteria are presented in the Fig. S1.

Only two carbon substrates, the fatty acid derivatives Tween 20 and Tween 40 were strongly positive for all of the ten MAH isolates. The kinetic curves for these substrates reached 250 Omnilog units, amongst the highest values recorded in our analysis (see Fig. S2 for all kinetic curves of the ten MAH isolates). The opm analysis revealed that a total of 20/379 (5.3%) substrates were metabolized differently among the MAH isolates (Table 3). We therefore carried out further analysis using only these substrates. The majority of these 20 substrates were carbon substrates, 15/20 (75.0%), followed by 3 nitrogen and 2 phosphorous substrates. The heatmap in Fig. 1 shows the utilization of these 20 substrates among the ten MAH isolates. The isolates are grouped according to their substrate utilization. Isolates utilizing similar substrates appear to cluster together.

Two major clusters, each composed of five isolates, could be observed. One was rich in environmental isolates (4/5) and the other was rich in clinical isolates (4/5). The

**Table 3  The 20 substrates differentiating the ten MAH isolates analyzed in this study.**

| PM Plate | Substrate and well number | Pathway involved | Reference |
|---|---|---|---|
| PM1 Carbon | Acetic acid –C08 | Pyruvate metabolism | *Baloni et al. (2014)*, *Nai et al. (2013)* |
| | Acetoacetic acid –G07 | Pyruvate metabolism | *Baloni et al. (2014)*, *Nai et al. (2013)* |
| | Methyl pyruvate –G10 | Pyruvate metabolism | *Baloni et al. (2014)*, *Nai et al. (2013)* |
| | Mono–methyl Succinate –G09 | Tricarboxylic acid cycle | *Baloni et al. (2014)*, *Nai et al. (2013)* |
| | Propionic acid –F07 | Propanoate metabolism, Nicotinate and nicotinamide metabolism, Degradation of aromatic compounds | *Baloni et al. (2014)*, *Kanehisa & Goto (2000)*, *Kanehisa et al. (2016)*, *Nai et al. (2013)* |
| | D-psicose –H05 | Glycolysis and branches | *Baloni et al. (2014)*, *Nai et al. (2013)* |
| | Pyruvic acid –H08 | Pyruvate metabolism | *Baloni et al. (2014)*, *Nai et al. (2013)* |
| | Tween 80 –E05 | Fatty acid metabolism | *Baloni et al. (2014)*, *Nai et al. (2013)* |
| PM2 Carbon | L-alaninamide –G02 | Amino acid metabolism | *Nai et al. (2013)* |
| | Butyric acid –D12 | Butanoate metabolism | *Baloni et al. (2014)*, *Kanehisa & Goto (2000)*, *Kanehisa et al. (2016)*, *Nai et al. (2013)* |
| | Caproic acid –E02 | Carboxylic acid metabolism | *Nai et al. (2013)* |
| | L-histidine –G06 | Amino acid metabolism | *Nai et al. (2013)* |
| | $\gamma$-hydroxy-butyric acid –E09 | Succinate metabolism | *Breitkreuz et al. (2003)*, *Nai et al. (2013)* |
| | $\beta$-methyl-D-galactoside –C07 | Galactose Metabolism | *Nai et al. (2013)* |
| | Sebacic acid –F08 | Carboxylic acid metabolism | *Nai et al. (2013)* |
| PM3 Nitrogen | D,L-$\alpha$-amino-caprylic acid –G10 | Amino acid metabolism | *Baloni et al. (2014)* |
| | L-cysteine –A11 | Amino acid metabolism | *Baloni et al. (2014)* |
| | D-galactosamine –E09 | Amino-sugar pathway | *Baloni et al. (2014)* |
| PM4 Phosphorous and sulphur | Carbamyl phosphate –B05 | Urea cycle and Pyrimidine synthesis | *Nelson & Cox (2004)* |
| | Sodium pyrophosphate –A03 | Phosphoric acid synthesis | *Nelson & Cox (2004)* |

substrates predominantly contributing to this clustering were butyric acid and propionic acid and indeed, the Tukey's test revealed that environmental isolates metabolized more strongly butyric acid ($p = 0.0209$) and propionic acid ($p = 0.00307$) than clinical isolates with statistical significance.

## Metabolic pathways analysis

The propionic and butyric acid are involved in three and one pathway, respectively (Table 3). A total of 151 genes have been identified in the KEGG database associated with all these pathways (*Kanehisa et al., 2016*). In Table S3 we reported the distribution and SNPs analysis of those genes in the MAH genomes. Of the 151 genes, 134 (88.7%) are present in all the ten MAH. The median gene length was 1,099 bp (range 318–2,253), whereas the median number of SNPs per gene is 17 (range 1–147). The phylogenetic analysis revealed that none of the SNPs could be associated with the group of the clinical or the group of environmental MAH isolates (see Table S4). In the propanoate pathway four operons have been identified: *fadAB* associated with the $\beta$-oxidation of several fatty acids (*DiRusso, 1990*), *ech8-9* encoding for hydrogenases that play a role in energy conversion (*Sant'Anna et al., 2015*), *sucCD* responsible for the succinate metabolism (*Cerdeno-Tarraga et al., 2003*) and *mutAB* involved in the methylmalonate pathway (*Schoenwolf*
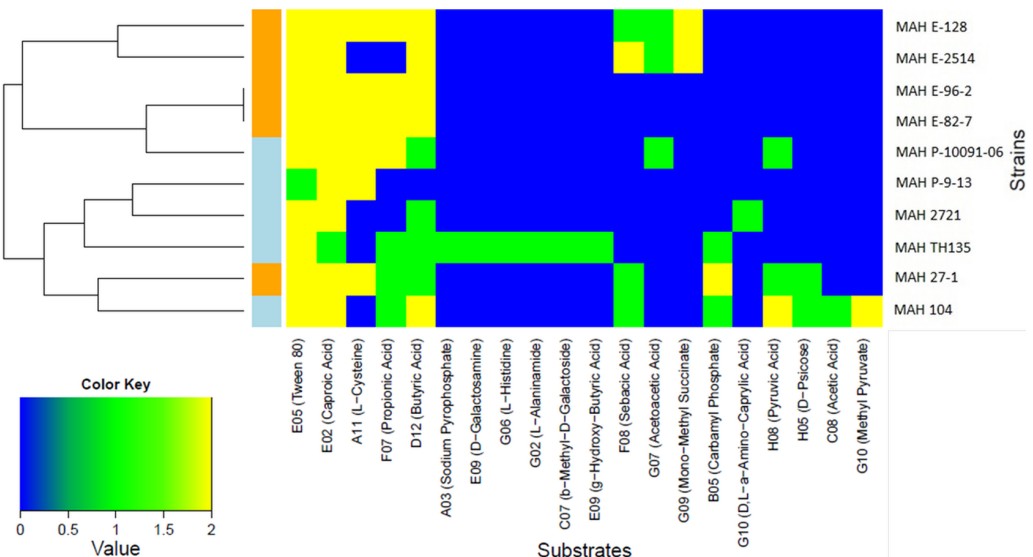

**Figure 1** **Heatmap showing the 20 substrates that were differently metabolized by the ten MAH isolates analyzed in this study.** The color key scale for each substrate is based on dye reduction quantified by Omnilog units. A yellow color indicates strong positive substrate metabolization, a green color moderate metabolization and a blue color indicates no substrate metabolization. Regarding the MAH isolates, environmental isolates are marked in orange, while clinical isolates are marked in blue.

& Alvarez, 1989). In the nicotinate pathway there are two operons: the *pntAA-AB-B* responsible for the transhydrogenation between NADH and NADP (*Anderlund et al., 1999*) and *nadABC* involved in the biosynthesis of NAD+ (*Vilcheze et al., 2010*). In the degradation of aromatic compounds pathway we identified the *pcaHGB* operon involved in the β-ketoadipate pathway (*Harwood & Parales, 1996*). In the butanoate pathway we identified the *fadAB* and *ech8-9operons*, the *sdhCDAB* encoding for the succinate dehydrogenase complex involved in the fatty acid metabolism (*Nam et al., 2005*) and the *ilvBN* responsible for the acetolactate synthesis, a precursor of several amino acids (*Keilhauer, Eggeling & Sahm, 1993*).

## Clustering analysis and determination of the accessory genome

The WGS of the two reference strains MAH 104 and MAH TH135 were already in the GenBank database and we submitted the remaining genomes at DDBJ/EMBL/GenBank under the BioProject Number PRJNA299461. The MCG, the maximum number of genes shared by all ten MAH isolates was 1,658, the alignment of which spanned 1.378 Mbp. The clustering analysis of the ten MAH isolates is shown in Fig. 2. By comparing the genetic clustering obtained by WGS with the phenotypic clustering obtained through BIOLOG PM we observed slight differences. For examples, the isolates MAH E-96-2 and MAH E-82-7, which share identical metabolic profiles, were genetically more distant from each other. Interestingly, at the genetic level there was no obvious clustering between the group of clinical and the group of environmental isolates.

The accessory genome is constituted by 4,067 genes. A total of 1,688 genes were specific for the group of clinical isolates (Table S5). On the other hand, 698 genes were specific

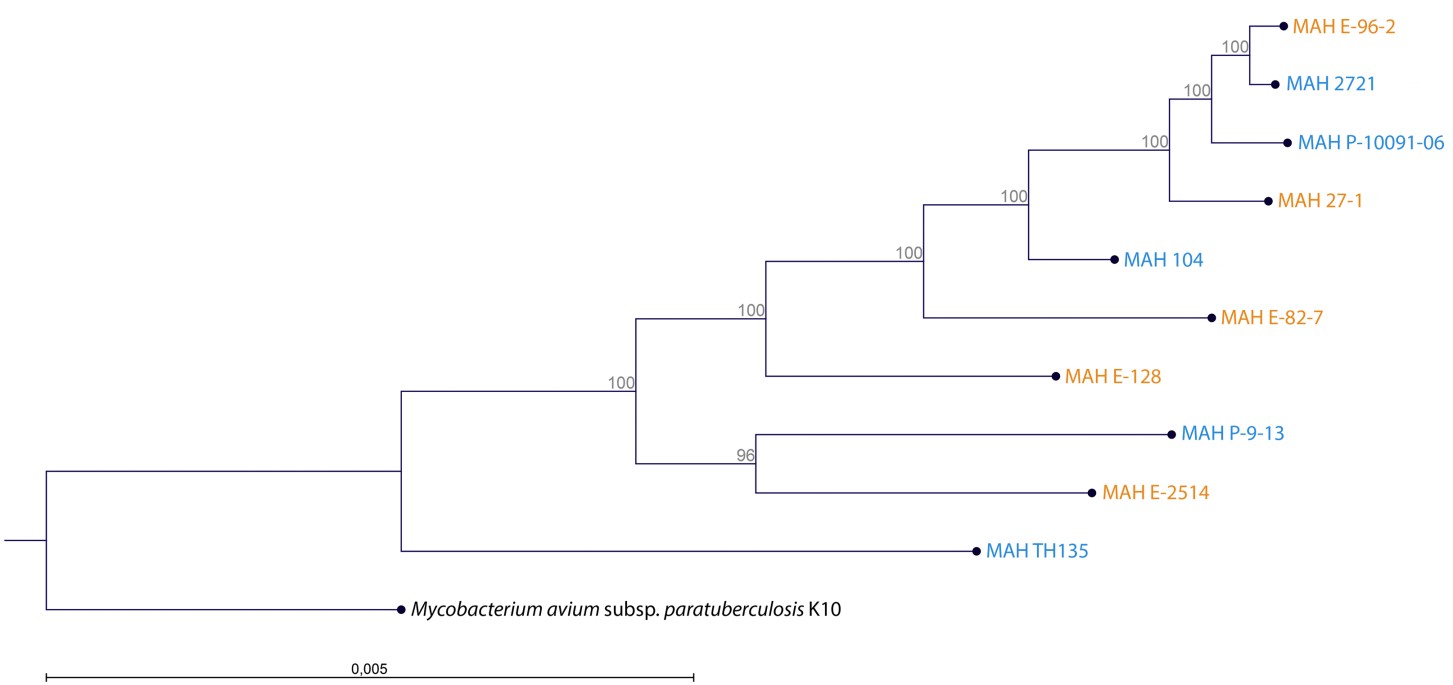

**Figure 2   Clustering of the 10 MAH isolates.** The tree was generated using RAxML 8.1. The alignment comprised 1,658 genes constituting the maximum common genome of our ten MAH isolates. Two reference strains were also included (MAH 104 and MAH TH135). The genome sequence of *M. avium* subsp. *paratuberculosis* K10 (Accession Number: AE016958) was used as outgroup. Isolate origin is also represented by blue for clinical origin and orange for environmental origin. The percentage of trees in which the associated taxa clustered together is shown adjacent to the branches.

for the group of environmental isolates (Table S6). We found no genes that were present in all the clinical and absent in all the environmental isolates, and vice-versa. Among the most abundant specific genes of the two groups of isolates, there were no known genes associated with the pathways which involved butyric and propionic acid. However, genes annotated as hypothetical proteins represented the most abundant specific genes of the two groups

## DISCUSSION

This study represents the first phenotypic analysis of a collection of clinical and environmental MAH isolates using the Biolog PM technology. We showed that the PM technology works well and can be performed with MAH isolates. Strong positive reactions with several substrates were observed with kinetic curves exceeding 200 Omnilog dye units. Although some substrates were metabolized only moderately by our MAH isolates (green in Fig. 1), this might be due to the fact that the use of such substrates by bacteria has a time lag.

The ten MAH isolates showed different metabolic patterns pointing to high intra-species diversity. Only two out of the ten isolates had identical heatmap profiles (MAH E-96-2 and MAH E-82-7).

Our study showed that MAH isolates prefer to metabolize fatty acids as a carbon source. Indeed, the Tween substrates were strongly metabolized by all MAH isolates

tested. This is in agreement with prior studies, showing that Tween substrates were widely used by different mycobacterial species (*Baloni et al., 2014*; *Chen, Scaria & Chang, 2012*; *Hayashi et al., 2010*; *Khatri et al., 2013*; *Lofthouse et al., 2013*; *Wang et al., 2011*). It has been reported that mycobacteria hydrolyze Tween 80 to generate the fatty acid oleic acid, which can enter the Tricarboxylic acid (TCA) cycle or can be used as a substrate for energy production (*Lofthouse et al., 2013*; *Vandal, Nathan & Ehrt, 2009*). Other fatty acids used by the majority of our MAH isolates are represented by two short fatty acids, caproic acid and butyric acid (*Kanehisa & Goto, 2000*; *Kanehisa et al., 2016*; *Khatri et al., 2013*). Caproic acid and its derivatives are involved in several mycobacterial pathways such as the degradation of aromatic compounds, oxocarboxylic acid metabolism or lysine degradation (*Kanehisa & Goto, 2000*; *Kanehisa et al., 2016*). The butyric acid is the final product of butanoate metabolism. Propionic acid is another fatty acid used by our MAH isolates and this represents the terminal product of propanoate metabolism (*Kanehisa & Goto, 2000*; *Kanehisa et al., 2016*). The nitrogen source L-cysteine, used by six of our MAH isolates, is the final product of cysteine metabolism and is involved in the biosynthesis of other amino acids such as methionine and histidine (*Baloni et al., 2014*; *Kanehisa & Goto, 2000*; *Kanehisa et al., 2016*).

The question of whether bacteria of the same species originating from either clinical or environmental sources differ from each other is still a matter of discussion. Li and co-authors (*2014*) showed that comparative genome analysis clearly distinguished clinical and environmental *Vibrio parahaemolyticus* isolates from each other. In contrast, other researchers have reported no difference between clinical and environmental *Pseudomonas aeruginosa* isolates with regard to virulence and metabolic properties (*Alonso, Rojo & Martinez, 1999*; *Vives-Florez & Garnica, 2006*). Although our study did not reveal any clear distinction between clinical or environmental MAH isolates at the level of the whole genome, we observed differences between clinical and environmental isolates with regard to substrate utilization. The most intriguing difference is that the two fatty acids butyric acid and propionic acid are metabolized more by the environmental than by clinical isolates.

We observed no difference in the presence / absence of genes associated with butyric or propionic acid pathways among the group of clinical and the group of environmental MAH isolates. The SNPs analysis of the genes involved in the pathways revealed that no SNPs were associated with clinical or environmental origin of the MAH isolates. These evidences suggest that the metabolic differences observed among clinical and environmental MAH isolates might be due to difference in gene regulation. However, we screened all the genes that up to now have been associated with the pathways of interest. We can speculate that there might be additional genes, of unknown function, that might play a role in the above pathways.

The analysis of the accessory genome revealed that none of the genes specific for the clinical or for the environmental isolates could be associated with the pathways of interest. However, future studies on the high number of hypothetical proteins might clarify whether they have a role in the pathways which involved the butyric and propionic acid.

The higher metabolic activity observed among environmental MAH isolates might be advantageous for survival in an environment presenting a wider range of nutritional conditions than the host cells alone. Further studies testing a larger number of isolates from different origins might clarify this. In addition, it has been showed that in bacteria the fatty acids have a role in adaptation to different environmental conditions (*De Sarrau et al., 2012*; *De Sarrau et al., 2013*; *Diomande et al., 2015*).

## CONCLUSIONS

Our study contributes to the understanding of the emerging pathogen MAH at the phenotypic and metabolic level. Understanding how bacteria utilize their own or host-derived substrates during infection might help the development of strategies to fight such infections. We encourage phenotypic testing of microbial isolates from different ecological niches to identify key substrates or pathways that can be used as targets for drug development or for selective growth media development.

## ACKNOWLEDGEMENTS

We would like to thank Barry Bochner (President of BIOLOG) for his invaluable input to this study and Brian Weinrick (Albert Einstein College of Medicine, New York City) for his support in developing the BIOLOG laboratory protocol. Elvira Richter (National Reference Center for Mycobacteria, Borstel, Germany) and Roland Schulze-Röbbecke (University Hospital Düsseldorf) provided a number of MAH isolates, Carsten Schwarz (Christiane Herzog Zentrum, Charité, Berlin) provided respiratory samples from cystic fibrosis patients and Kei-ichi Uchiya provided the reference strain MAH TH135. We thank Katharina Schaufler (Free University Berlin) for her support with the BIOLOG data analysis and Inga Eichorn (Free University Berlin) for her help with the whole genome sequencing data. We thank Steve Norley (Robert Koch Institute, Berlin) for the English revision of the manuscript.

### Funding

This work was supported by a grant from the German Research Foundation (DFG)-sponsored International Research Training Group (IRTG) entitled 'Internationales Graduiertenkolleg –Functional Molecular Infection Epidemiology –GRK1673 (Berlin-Hyderabad)' to AS and FD. The funders had no role in study design, data collection and analysis, decision to publish, or preparation of the manuscript.

### Grant Disclosures

The following grant information was disclosed by the authors:
German Research Foundation (DFG)-sponsored International Research Training Group (IRTG).

## Competing Interests

The authors declare there are no competing interests.

## Author Contributions

- Andrea Sanchini performed the experiments, analyzed the data, wrote the paper, prepared figures and/or tables, reviewed drafts of the paper.
- Flavia Dematheis conceived and designed the experiments, performed the experiments, analyzed the data, contributed reagents/materials/analysis tools, prepared figures and/or tables, reviewed drafts of the paper.
- Torsten Semmler analyzed the data, contributed reagents/materials/analysis tools, prepared figures and/or tables, reviewed drafts of the paper.
- Astrid Lewin conceived and designed the experiments, analyzed the data, wrote the paper, reviewed drafts of the paper.

## Data Availability

The raw data has been supplied as Supplementary Files.

## Supplemental Information

Supplemental information for this article can be found online at http://dx.doi.org/10.7717/peerj.2833#supplemental-information.

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
