# Peer review of "Metabolic phenotype of clinical and environmental Mycobacterium avium subsp. hominissuis isolates"

_PeerJ, doi:10.7717/peerj.2833_

## Round 0.1 · original submission · Major Revisions

The underlining experiments are rather simple and straightforward, but I believe they were completed without issues and generated valid sets of data. However, I have to agree with the reviewer 1 that the subsequent bioinformatics analysis still needs some work and is strongly based on the similar analysis performed in the reference article. More analyses are needed, e.g. the comparison with other species. As a result, the conclusions are weak, even if this great work (especially the collected data) deserves a much better outcome. I believe there is still an opportunity to redo this work in a more elaborate way and to arrive at the results worthy of publication as long as the authors can correct their bioinformatics shortcomings and rerun the analysis on the data they collected. Then it would be an excellent piece of work.

Reviewer 1 ·

Basic reporting

The paper is structured appropriately per specified PeerJ policies. It is written in clear English, and requires only minor grammatical changes and/or typo corrections.

Experimental design

The experimental part of the paper is very basic and straight-forward, only one experiment was performed.
It is not clear from the text (Material and Methods, page3) what the parameters for screening of genes of interest in ten MAH genomes are and that topic needs to be explicitly explained. Additionally, how did the authors selected orthologs and how were they different from paralogs in their analysis?

Validity of the findings

In the result part of the paper authors claimed:
“Regarding the analysis of the metabolic pathways of these two substrates, in the reference strain MAH 104, 147 genes are known to be involved in the butyric acid and propionic acid pathways (Kanehisa et al. 2016) Based on WGS of the ten isolates we analyzed the distribution of these genes in our ten isolates. However, no difference was observed in gene distribution among clinical and environmental MAH isolates (data not shown), pointing to difference in gene regulation and not in gene content.”
As I believe myself to be a strong supporter of the importance of bacterial gene regulation, I think the above statement is not exactly corollary of the analysis performed. The only thing that can be claimed for certain is that it is unlikely that the differences in the phenotypes can be explained by knock-out of one of 147 genes which were known to be involved in the butyric and propionic acids pathways in MAH 104 . (Note that this was the case for a leuD mutant of the M.avium substr paratuberculosis described in Chen et al. 2012). However, it is possible that some other genes or operons involved in the butyric and propionic acids pathways are present in some or even all isolates, but not in MAH 104.
Moreover, according to the paper’s findings, environmental isolates were more metabolically active than clinical isolates – adding further validity to the idea of presence of “environmental specific” genes that are coding proteins involved in the metabolism of fatty acids.
While it is still very possible that the different metabolic activities of these isolates are a result of different regulation, I believe that a more accurate genomics analysis should be performed to establish certainly. In addition to the analysis of common genes the separate analysis of the clinical/environmental specific genes should also be completed.
Furthermore (and in contrast of focusing on simple presence/absence of orthologs genes in the isolates), the analysis of potential operon-duplication, operon re-arrangements and even gene similarity should be performed.
Finally, in the “Clustering analysis by whole genome sequencing” section of the results, the authors provide a maximum number of genes shared by ten isolates. I think it would be very interesting for other researchers to know the answers to the following related questions, such as: how many shared operons these ten isolates have, how many genes/operons are unique for each isolate, what is a number of shared genes/operons per clinical, environmental isolates, some other clusters of authors’ interest, etc.

Additional comments

In the beginning of the introduction the authors start with: "In recent years, there have been substantial advances in the analysis of bacteria at the molecular level...In contrast, there has been little concomitant advance in knowledge at the phenotypic level. Phenotype analysis deserves greater attention, as it is the phenotype that selection pressure acts upon to confer evolutionary advantages to the bacteria species".
The paper’s introduction is completed with the following stated aim:
"Our aim was to investigate if the PM technology was applicable to MAH isolates, to describe the metabolic substrates utilized by MAH isolates and to identify any metabolic differences between clinical and environmental MAH isolates"
I believe it to be an aim of self-limiting ambition for a number of reasons. First of all, PM technology was already shown working successfully with a number of Mycobacterium species such as M. tuberculosis, M bovis, M.smegmatis and even M.avium subsp paratuberculosis, which makes experimental part of the paper rather ordinary and straightforward. Second, when in possession of both phenotypic and genomics data I would expect to see a detailed combined analysis. Unfortunately that didn’t happen, while the genomics analysis part was done poorly and the summary was done inaccurately.
I hope that the authors will re-focus their efforts on performing the combined analysis of the genetics and phenotype data, especially keeping in mind that all data sets have already been generated and are available.

·

Basic reporting

The authors have described this study using a clear English language.
The introduction of the story is quite complete with many interesting references.
If it is possible I would suggest to improve the figure 2 while the remaining figures and tables are well labelled and described.

Experimental design

The experimental design is very simple and for this reason I don't have any negative comments.
It was a good idea from the authors to use microarray technology in order to identify which substrate are metabolized by MAH isolates. I think that this method can also be applied to other bacteria strains.

Validity of the findings

no comments

---

## Round 0.2 · Minor Revisions

I highly appreciate changes so quickly made to the manuscript.
However, I would be grateful for improvements made according to Reviewer 1 suggestions. Also I'm sad authors did not include more distant species into analyses which would broaden the audience interested in this work. I do not however consider this job as a must but I think it is a way to put your own work into a wider context.

Reviewer 1 ·

Basic reporting

Clear, unambiguous, professional English language used throughout.

Experimental design

The experimental part the of the paper is very basic and straight-forward.

Validity of the findings

Authors have accounted for the majority of the suggestions to improve the quality of the paper.
In the "Metabolic pathway analysis" they wrote : "The phylogenetic analysis revealed that none of the SNPs could be associated with the group of the clinical or the the group of environmental MAH isolates." It is not in fact clear from the text how this analysis was performed or were the phylogenetic trees build using nucleotide or amino acid sequences.
It also not clear if the calculated SNPs are all SNPs or non-synonymous only, or if it is a combination of syn and non-syn SNPs. It would be nice to see how many of them were non-synonymous and frame-shift.
Finally, even if the phylogenetic analysis didn't demonstrate any association between SNPs and group of isolates, it would be helpful to have these trees in supplementary materials.

·

Basic reporting

no comments

Experimental design

no comments

Validity of the findings

no comments

Additional comments

good job!

---

## Round 0.3 · accepted · Accept

Dear Astrid,

Thank you very much for your quick reply and work done. I think that even with a few flaws left (e.g. the phylogenetic analysis realized on gene sequences only) it fully deserves to be read by experts in the field. Good luck! :-)

Best wishes,
Marcin